# Retinopathy of Prematurity and Hearing Impairment in Infants Born with Very-Low-Birth-Weight: Analysis of a Korean Neonatal Network Database

**DOI:** 10.3390/jcm10204781

**Published:** 2021-10-19

**Authors:** Chang Myeon Song, Ja-Hye Ahn, Jae Kyoon Hwang, Chang-Ryul Kim, Mi Jung Kim, Kyeong Mi Lee, Hyun Ju Lee, Seong Joon Ahn

**Affiliations:** 1Department of Otorhinolaryngology-Head & Neck Surgery, Hanyang University Hospital, Hanyang University College of Medicine, Seoul 04763, Korea; cmsong@hanyang.ac.kr; 2Division of Neonatology and Developmental Medicine, Hanyang University Hospital, Seoul 04763, Korea; mdscully@gmail.com (J.-H.A.); leek0219@naver.com (K.M.L.); 3Department of Pediatrics, Hanyang University Hospital, Hanyang University College of Medicine, Seoul 04763, Korea; 4Department of Pediatrics, Hanyang University Guri Hospital, Hanyang University College of Medicine, Guri 11923, Korea; fanta1412@naver.com (J.K.H.); crkim@hanyang.ac.kr (C.-R.K.); 5Department of Rehabilitation Medicine, Hanyang University College of Medicine, Seoul 04763, Korea; kimmjreh@hanyang.ac.kr; 6Department of Ophthalmology, Hanyang University Hospital, Hanyang University College of Medicine, Seoul 04763, Korea

**Keywords:** retinopathy of prematurity, hearing impairment, screening, very low birth weight

## Abstract

Background: To investigate hearing impairment and its association with retinopathy of prematurity (ROP) among children born with very low birth weight (VLBW, birth weight < 1500 g). Methods: This prospective registry study included 7940 VLBW infants who underwent both ophthalmic (ROP) and hearing screening at the 70 participating centers of the Korean Neonatal Network. Hearing screening was performed using auditory brainstem response and/or automated otoacoustic emission testing. Hearing impairment, defined as a unilateral or bilateral hearing threshold of ≥40 dB on the auditory brainstem response threshold (ABR-T) test, was evaluated and compared between children with and without ROP at the corrected ages of 18 months and 3 years. Results: The frequency of infants who did not undergo hearing screening at near-term ages was higher in the ROP group than in the no-ROP group (18.2% vs. 12.0%, *p* < 0.001), and the prevalence of hearing impairment at 18 months was higher in the ROP group than in the no-ROP group (3.5% vs. 2.2%, *p* = 0.043). The prevalence of deafness was higher in children with ROP than those without ROP (0.4% vs. 0.1%, *p* = 0.049). There were significant differences in hearing impairment among the stages of ROP (*p* < 0.001). However, multivariate analyses and propensity score matching showed no significant association between ROP and hearing impairment at 18 months and 3 years after adjusting for prematurity-related variables (all *p* > 0.05). Conclusions: Among infants born with VLBW, hearing impairment was more common in those with ROP than in those without ROP at 18 months of age. However, there was no significant independent association between hearing impairment and ROP.

## 1. Introduction

Retinopathy of prematurity (ROP) is a retinal vasoproliferative disease mediated by vascular endothelial growth factor (VEGF), which may threaten vision in preterm infants [1,2]. Despite advances in neonatal intensive care, ROP remains one of the leading causes of childhood blindness. Vision impairment due to ROP can be a significant burden to children, families, and society as preterm births increase and the number of those at risk of ROP has been increasing worldwide [1,2].

In addition to the visual disabilities directly caused by ROP-related complications, hearing impairment may also affect the infants with ROP as preterm infants are at risk of major neurosensory impairments, including visual and hearing ones [3]. Both impairments may significantly disturb child development because these two senses are vital for the development of other domains such as the cognitive, linguistic, and social-emotional ones [4,5]. However, hearing impairment has not been extensively addressed among infants with ROP, although hearing is the main neurosensory domain that visually deprived children with ROP rely on for their development.

In the present study, we aimed to investigate hearing impairment in infants with and without ROP in a large prospective cohort of infants born with very low birth weight (VLBW, birth weight < 1500 g). In this cohort consisting of Korean VLBW infants with ophthalmic and hearing assessment, we intended to explore the relationship between ROP and hearing impairment, as well as between the two types of sensory impairment.

## 2. Materials and Methods

### 2.1. Study Population

The Korean Neonatal Network (KNN) is a prospective national registry of infants with VLBW, which was established in 2013 by 70 participating hospitals with governmental support from the Korea Centers for Disease Control and Prevention. The registry included clinical data on ROP details and systemic conditions, prenatal/perinatal factors, maternal factors, systemic or ROP management, and hearing assessments. Among the 10,424 infants enrolled between January 2013 and December 2017 in the registry, 1803 were excluded because they had not undergone ophthalmic examination, mainly due to death associated with preterm birth-related complications (*n* = 1462). Among the 8621 infants with VLBW, those with hearing outcomes evaluated using well-established neonatal screening methods (*n* = 7940) were included in our analyses. Among them, the hearing outcomes assessed in 2863 infants at 18 months or 3 years of age were analyzed using univariate or multivariate analyses. Figure 1 presents a flowchart of the participant inclusion and exclusion criteria in our study cohort. The present study was performed in accordance with the Declaration of Helsinki and was approved by the KNN data management committee and the Hanyang University Hospital IRB (Protocol No. 2021-02-011).

### 2.2. Ophthalmic and Hearing Assessments and Patient Classification

Ophthalmic screening for ROP followed an established guideline set by several pediatric and ophthalmologic organizations, which recommends screening for ROP in infants born with body weights less than 1500 g or gestational age (GA) less than 30 weeks [6]. All included infants with VLBW underwent binocular indirect ophthalmoscopy performed by experienced ophthalmologists. The ROP stages were classified from 1 to 5 according to the International Classification of Retinopathy of Prematurity [7,8,9,10]. Detailed information on ROP, including treatment details (i.e., surgery and anti-VEGF therapy) and maximal stage of ROP, were also retrieved from the database. The treatment for ROP was performed by well-trained ophthalmologists in the participating centers.

For hearing screening, we used automated auditory brainstem response and/or automated otoacoustic emission testing at near-term ages [11]. If newborns failed the hearing screening in either ear, they were referred to the otolaryngology department for diagnostic auditory brainstem response threshold (ABR-T) testing and/or impedance audiometry and otoacoustic emission testing. For the children who had been admitted to the neonatal intensive care unit for ≥5 days, a hearing assessment was performed every 6 months or 1 year, regardless of the results of the screening, to diagnose delayed hearing loss. Classification of hearing loss was defined based on the protocol of the KNN Database, according to ABR-T. Hearing impairment was defined as a unilateral or bilateral hearing threshold of ≥40 dB. Mild, moderate, and severe hearing loss and deafness were defined as thresholds of ≥40 dB, ≥50 dB, ≥70 dB, and ≥90 dB HL, respectively. Unilateral or bilateral use of hearing aids or cochlear implantation was evaluated on the visit to the outpatient clinic at 18 months and 3 years of corrected age.

We divided the infants into groups based on the presence of ROP (without ROP and with ROP). The treated infants were further separated according to treatment modalities, including laser photocoagulation alone, anti-VEGF therapy alone, and both. Visual impairment was defined as unilateral or bilateral blindness based on examination by an ophthalmologist.

### 2.3. Data Analysis

Data on maternal characteristics, neonatal characteristics, and ROP details were collected from the registry. Patients with comorbidities, including necrotizing enterocolitis (NEC, staged according to modified Bell’s staging criteria), cerebral palsy (diagnosed with a structured neurological examination), and bronchopulmonary dysplasia (BPD, severity determined by the physiologic definition) [12,13], were also used for our analyses.

We compared the maternal and neonatal clinical characteristics and comorbidities of prematurity between infants with and without ROP, as well as between the treatment groups. We also compared the hearing outcomes between groups separated based on the presence of ROP and the treatment modalities, including laser photocoagulation, anti-VEGF therapy, and both laser and anti-VEGF therapy. Propensity score matching was performed including perinatal demographics such as GA and birthweight and clinical characteristics such as intraventricular hemorrhage (IVH), clinical sepsis, and BPD, and infants were divided into prematurity-matched groups with and without ROP. A chi-squared test or Fisher’s exact test for categorical variables and a Student’s t-test or analysis of variance (ANOVA) for continuous variables were used to compare groups. Multivariate regression was used to analyze the associations between hearing outcomes and ROP, as well as those between the outcomes and treatment modalities, by adjusting for other confounding variables including GA, sex, BPD, IVH, clinical sepsis, and periventricular leukomalacia (PVL). Values for continuous variables are presented as the mean ± standard deviation. Statistical analyses were performed using SPSS software (version 26.0; SPSS, Chicago, IL, USA). Statistical significance was set at *p* < 0.05.

## 3. Results

### 3.1. Demographic and Clinical Characteristics

Of the 7940 infants who underwent hearing and ophthalmic screening, 2480 (31.2%) had ROP. Among the 2863 infants with both ophthalmic and hearing examinations at follow-up visits, 982 infants (34.3%) had ROP, consisting of 305 (31.1% of ROP infants), 336 (34.2%), 336 (34.2%), 5 (0.5%), and 0 with maximal stages 1, 2, 3, 4, and 5, respectively. The outcomes in terms of ROP and functional (visual/hearing) impairment among the infants receiving both hearing and ophthalmic assessments are summarized in Appendix A.

The demographic and clinical characteristics of the infants with and without ROP among those with hearing outcomes at follow-up visits are presented in Table 1. The mean GA was 29.6 ± 2.4 in infants without ROP and 26.4 ± 2.1 weeks in those with ROP, which was significantly different. The birth weights of infants with and without ROP were 907.6 ± 247.3 g and 1191.2 ± 218.3 g, respectively (*p* < 0.001). Several maternal characteristics and neonatal comorbidities, including pregnancy-induced hypertension, chorioamnionitis, respiratory distress syndrome, sepsis, necrotizing enterocolitis, BPD, IVH, and patent ductus arteriosus ligation, were significantly different between infants with and without ROP. However, the differences in GA, birth weights, and other comorbidities between the groups were remarkably reduced between the two groups with propensity score matching (Table 1).

Approximately one-third of infants with ROP (334, 34.0%) received ROP treatment, including laser photocoagulation alone (219, 65.6% among those with treated ROP), anti-VEGF therapy alone (63, 18.9%), or both treatments (52, 15.6%).

### 3.2. Hearing Outcomes

Among the 7940 infants who underwent both ROP and hearing screening, 452 of 2480 infants with ROP failed to pass the hearing screening, whereas 657 of 5460 infants without ROP did. Accordingly, the prevalence of those who did not pass the hearing screening was 18.2% and 12.0% among the infants with and without ROP, respectively, which was significantly different.

Among the 2863 children with a hearing assessment at follow-up visits, 2751 and 666 received the assessment at the first and second follow-up visits, respectively. Hearing impairment was noted in 73 and 12 infants at the first and second visits, respectively, resulting in a 2.7% and 1.8% prevalence of hearing impairment, respectively. Bilateral hearing impairment was noted in 35 (1.3%) and 5 (0.8%) infants, and deafness was noted in 5 (0.2%) and 2 (0.3%) infants at the first and second visits, respectively.

Table 2 summarizes the hearing outcomes of patients with and without ROP. The prevalence of hearing impairment among infants with ROP at the first visit was 3.5% (33 out of 939), whereas that among those without ROP was 2.2% (40 of 1812). There was a significant difference in the frequency of hearing impairment between those with and without ROP. The frequency of deafness was significantly greater in patients with ROP than in those without ROP (0.4% vs. 0.1%). At the second visit, however, there were no significant differences in the frequencies of hearing impairment or deafness between the groups. There were no significant differences in hearing aid use or cochlear implantation for hearing impairment between the two groups at either visit. With propensity score matching, there were no significant differences in any hearing outcome between the infants with and without ROP.

Figure 2 shows the prevalence of hearing impairment in subgroups divided by the ROP stage. The prevalence of hearing impairment at 18 months showed an increasing tendency with advancing stages of ROP; in particular, the prevalence of hearing impairment in infants with stage 4 ROP was 40%. The difference in the prevalence at the first follow-up visit was significantly different between the different stages. However, at 3 years of age, there were no significant differences in the prevalence between ROP stages, which were not statistically significant.

Appendix A shows the hearing outcomes in the subgroups that were separated based on the treatment modalities of ROP. The prevalence of hearing impairment was 5.1%, 1.6%, and 7.7% among infants treated with anti-VEGF, laser photocoagulation, and both, respectively. The differences between the treatment groups were not statistically significant.

### 3.3. Association between Hearing Impairment and ROP

Table 3 shows the clinical factors associated with hearing impairment in the regression analyses. Univariate analyses revealed a significant association between hearing impairment at 18 months and GA, sex, ROP, IVH, BPD, and sepsis. However, the association at 3 years was only significantly associated with GA and sepsis.

In multivariate analyses, however, ROP showed no significant association with hearing impairment at the first or second follow-up visit. The clinical factors associated with that at the first and second visit were IVH (*p* = 0.015) and GA (*p* = 0.038), respectively, with the odds ratios of 1.30 (95% confidence interval (CI), 1.05–1.60) and 0.72 (95% CI, 0.53–0.98), respectively. Accordingly, after adjusting for other demographic- or prematurity-associated factors, hearing impairment was not significantly associated with ROP at 18 months or 3 years of age.

Appendix A shows the prevalence of hearing impairment in individuals with and without visual impairment. At 18 months of age, hearing impairment was noted in 7.7% and 2.6% of infants with and without visual impairment, respectively. At both 18-month and 3-year visits, the differences in prevalence were not statistically significant. Furthermore, logistic regression analyses showed that hearing impairment was not associated with visual impairment after adjusting for GA, sex, and sepsis (*p* = 0.910 and 0.583 at the first and second visits, respectively).

## 4. Discussion

Using the data of a large number of VLBW infants registered in a nationwide cohort, our study revealed the frequency of both ROP and hearing impairment and explored the association between them. To the best of our knowledge, this is the first longitudinal cohort study to evaluate the relationship between ROP and hearing impairment.

With hearing outcomes obtained longitudinally at near-term, and 18 months and 3 years after birth, the present study suggests that the infants born with VLBW and also ROP may be at risk of hearing impairment at early developmental periods. However, hearing development may be affected by multiple clinical and prematurity-related factors, and adjustment for the factors, either multivariate regression analyses or propensity score matching, resulted in a non-significant association between hearing impairment and ROP or visual impairment.

Preterm birth is known to increase the risk of neurodevelopmental delay and neurosensory impairment [14]. Further, major medical conditions associated with prematurity can also affect children’s neurodevelopmental outcomes in the cognitive, linguistic, and motor domains [14]. For example, ROP has been reported to be associated with poor neurodevelopmental outcomes, although conflicting results have also been reported after adjusting for prematurity and comorbid conditions [15,16]. Hearing outcomes may also be affected by preterm birth as well as prematurity-associated conditions including ROP [3,12,13], as other neurodevelopmental outcomes were.

A national register study of preterm infants performed in Finland showed that the overall prevalence of hearing loss among infants with GA < 32 weeks was 2.46%, similar to 2.65% in our study, and hearing impairment decreased with advancing gestational age [3]. Furthermore, in this study, GA was significantly associated with hearing impairment in univariate or multivariate analyses, as shown in Table 3. This suggests that (extreme) prematurity is one of the risk factors for hearing impairment.

Stadio et al. showed that in newborns who were hospitalized in neonatal intensive care units, the incidence of hearing loss was increased in those with retinopathy compared to that in those without [17]. In this study, hearing impairment was diagnosed in 3.5% and 2.1% of the children with ROP at 18 months and 3 years of age, respectively, which were greater than those without ROP. Schmidt et al. showed a higher prevalence of severe hearing loss in those with severe ROP (13.2%) than in those without (2.4%) [18], which was similar to our findings presented in Figure 2, showing a greater prevalence of hearing impairment in those with stage 4 ROP. However, whereas the prevalence of hearing impairment and deafness at 18 months was significantly higher in infants with ROP than in those without ROP, hearing impairment was not significantly different at 3 years of age. These results suggest the possible association between ROP and hearing impairment as well as delayed auditory neurodevelopment [19] in infants with ROP, which requires further validation.

However, hearing is clinically important for children with ROP as visually deprived infants with ROP should rely on auditory perception for their development. Hearing loss and ROP may share a common pathogenic background or be affected by various factors such as prematurity, mechanical ventilation, sepsis, and neurological disorders [20,21,22]. Therefore, the effects of other confounders should be carefully excluded to validate the true association between ROP and hearing impairment. However, the association has rarely been addressed or carefully examined by the adjustment of other prematurity-associated factors. In the present study, multivariate analyses or propensity score matching showed no significant associations between ROP and hearing impairment at 18 months or 3 years after birth. This indicates that ROP may not independently affect hearing development, although those with ROP may be at risk of hearing impairment during the early developmental period. From the association between GA and hearing impairment, the greater prevalence of hearing impairment in children with ROP might be associated with extreme prematurity in those with ROP.

Risk factors for ROP suggested by previous studies include maternal factors (maternal diabetes mellitus and hypertension), prenatal and perinatal factors (chorioamnionitis and premature rupture of membrane), infant factors (low gestational age [23], small birth weight [23], twin birth [24], and low Apgar score [25]), medical interventions (oxygen [26], respiratory support [27]), comorbidities of prematurity (BPD [28], IVH [29], NEC [30], and sepsis [29]), and genetic factors [31,32]. Those of childhood hearing loss contain preterm birth, intracranial hemorrhage, the use of ototoxic medications, congenital cytomegalovirus, toxoplasmosis, admission to a NICU, and a family history of hearing loss [3,33]. Extreme prematurity and associated comorbidities were the common risk factors for both ROP and hearing loss. Therefore, after controlling for extreme prematurity, the association between ROP and hearing loss was shown to be nonsignificant, and, mechanistically, the functional development of hearing might not be directly related to the pathologic processes (i.e., retinal vasoproliferation) occurring in ROP.

Several limitations of this study warrant careful consideration when interpreting the results. First, although this study was performed in a prospective manner, selection bias is likely to be present as hearing assessments were performed at follow-up visits in only 2863 of 10,424 infants. In particular, only 666 patients with available data on ROP examination received hearing assessments at the second visit, which might have resulted in significant selection bias. This was due to a follow-up loss of a significant portion of the infants at subsequent visits [34], which makes it impossible to draw conclusions regarding the long-term associations between ROP and hearing impairment. Clinical practice in ROP examination and treatment varied between participating centers; for example, some clinicians preferred anti-VEGF therapy, whereas others treated their patients with laser photocoagulation. This may lead to bias, although there were no significant associations between treatment modality and hearing outcomes.

## 5. Conclusions

Among VLBW infants, the frequency of hearing impairment at 18 months of age was higher in those with ROP than in those without ROP; however, ROP was not independently associated with hearing impairment after adjusting for prematurity and other clinical factors. Furthermore, visual impairment was not significantly associated with hearing impairment in children born with VLBW. This might suggest different pathogenic mechanisms between ROP and hearing impairments.

## Figures and Tables

**Figure 1 jcm-10-04781-f001:**
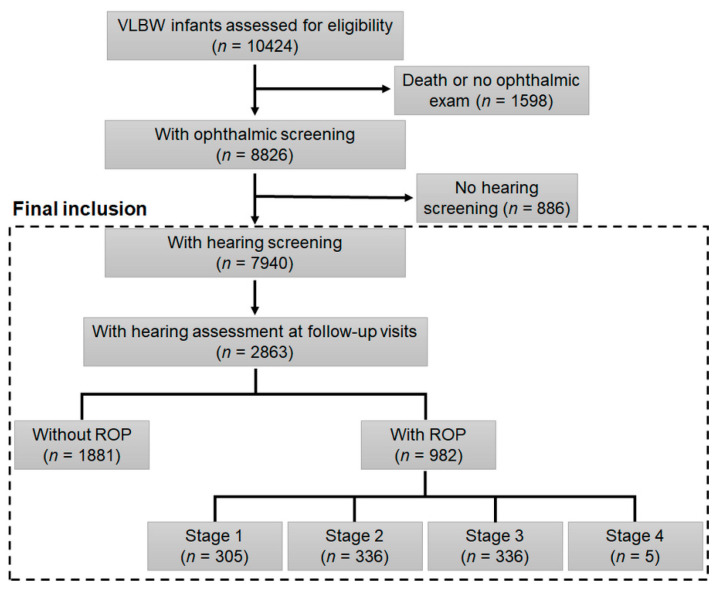
Flowchart illustrating the inclusion and exclusion criteria of the present study and the number of infants meeting the criteria. VLBW = very Low birthweight; ROP = retinopathy of prematurity.

**Figure 2 jcm-10-04781-f002:**
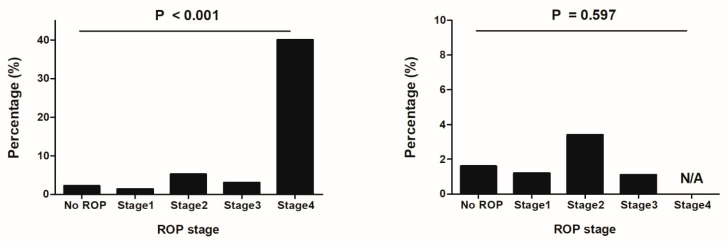
Prevalence of hearing impairment in subgroups of retinopathy of prematurity stages at 18 months (**left**) and 3 years (**right**) of age.

**Table 1 jcm-10-04781-t001:** Demographic and clinical characteristics of premature infants with and without retinopathy of prematurity among those who received ophthalmic and hearing assessment at follow-up visits.

Characteristics	All Infants			Propensity Score-Matched Infants		
	Without ROP (*n* = 1881)	With ROP (*n* = 982)	*p*-Value *	Without ROP (*n* = 626)	With ROP (*n* = 626)	*p*-Value *
Maternal characteristics						
Maternal PIH	470/1881 (25.0%)	136/982 (13.8%)	0.073	118/626 (18.8%)	101/626 (16.1%)	0.206
Maternal steroid use	1434/1860 (77.1%)	783/966 (81.1%)	0.027	503/615 (81.8%)	491/615 (79.8%)	0.385
Chorioamnionitis	476/1586 (30.0%)	348/843 (41.3%)	<0.001	195/535 (36.4%)	199/538 (37.0%)	0.854
Maternal PROM	623/1872 (33.3%)	401/973 (41.2%)	<0.001	234/622 (37.6%)	251/621 (40.4%)	0.312
Maternal GDM	159/1881 (8.5%)	61/982 (6.2%)	<0.001	41/626 (6.5%)	52/626 (8.3%)	0.236
Low maternal education level ^†^	17/1585 (23.6%)	10/843 (22.3%)	0.840	6/530 (1.1%)	9/533 (1.7%)	0.442
Neonatal characteristics and comorbidities						
Male sex	954/1881 (50.7%)	495/982 (50.4%)	0.875	321/626 (51.3%)	316/626 (50.5%)	0.777
Gestational age, mean (SD), weeks	29.6 ± 2.4	26.4 ± 2.1	<0.001	27.7 ± 1.9	27.4 ± 2.0	0.009
Birth weight, mean (SD), g	1191.2 ± 218.3	907.6 ± 247.3	<0.001	1029.9 ± 212.1	1006.4 ± 236.4	0.064
RDS	1348/1881 (71.7%)	929/982 (94.6%)	<0.001	566/626 (90.4%)	578/626 (92.3%)	0.227
BPD	850/1872 (45.4%)	849/981 (86.5%)	<0.001	488/623 (78.3%)	495/625 (79.2%)	0.707
PDA ligation	109/1870 (5.8%)	210/948 (22.2%)	<0.001	73/622 (11.7%)	92/608 (15.1%)	0.081
Sepsis	262/1881 (13.9%)	290/982 (29.5%)	<0.001	141/626 (22.5%)	151/626 (24.1%)	0.504
NEC stage ≥ 2	46/1880 (2.4%)	87/982 (8.9%)	<0.001	25/626 (4.0%)	45/626 (7.2%)	0.014
IVH grade ≥ 3	62/1880 (3.3%)	123/982 (12.5%)	<0.001	43/626 (6.9%)	51/626 (8.1%)	0.391
Postnatal steroid use	254/1881 (13.5%)	502/982 (51.1%)	<0.001	158/626 (25.2%)	253/626 (40.4%)	<0.001
PVL	95/1880 (5.1%)	116/981 (11.8%)	<0.001	36/626 (5.8%)	62/626 (9.9%)	0.006

Abbreviations: ROP, retinopathy of prematurity; PIH, pregnancy-induced hypertension; PROM, premature rupture of membranes; GDM, gestational diabetes mellitus; RDS, respiratory distress syndrome; BPD, bronchopulmonary dysplasia; NEC, necrotizing enterocolitis; IVH, intraventricular hemorrhage; PDA, patent ductus arteriosus; PVL, periventricular leukomalacia. * For categorical variables, the chi-squared test was used to compare groups. For continuous variables, the unpaired t-test was used to compare groups. ^†^ Mothers with fewer than 12 years of education.

**Table 2 jcm-10-04781-t002:** Hearing impairment in infants with and without retinopathy of prematurity (ROP) at 18 months (first follow-up) and 3 years of age (second follow-up).

Hearing Outcomes	All Infants						Propensity Score-Matched Infants					
	Month 18			Year 3			Month 18			Year 3		
	Without ROP (*n* = 1812)	With ROP (*n* = 939)	*p*-Value *	Without ROP (*n* = 421)	With ROP (*n* = 234)	*p*-Value *	Without ROP (*n* = 603)	With ROP (*n* = 606)	*p*-Value *	Without ROP (*n* = 166)	With ROP (*n* = 169)	*p*-Value *
**Overall impairment**	40 (2.2%)	33 (3.5%)	0.043	8 (1.9%)	5 (2.1%)	0.665	22 (3.6%)	18 (3.0%)	0.510	3 (1.8%)	1 (0.6%)	0.368
**Severity ^†^ **												
Mild	15 (0.8%)	9 (1%)	0.727	0 (0%)	2 (0.9%)	0.127	6 (1.0%)	4 (0.7%)	0.546	0	0	1.000
Moderate	5 (0.3%)	9 (1%)	0.017	2 (0.5%)	1 (0.4%)	1.000	5 (0.8%)	3 (0.5%)	0.506	0	0	1.000
Severe	12 (0.7%)	4 (0.4%)	0.599	4 (1.0%)	1 (0.4%)	0.660	8 (1.3%)	3 (0.5%)	0.143	2 (1.2%)	0	0.245
Deafness	1 (0.1%)	4 (0.4%)	0.049	1 (0.2%)	1 (0.4%)	1.000	0	4 (0.7%)	0.124	1 (0.6%)	1 (0.6%)	1.000
**Laterality, bilateral**	21 (52.5%)	14 (42.4%)	0.391	2 (25%)	3 (60%)	0.293	10 (1.7%)	7 (1.2%)	0.457	0	1 (0.6%)	1.000
**Hearing aid use**	9 (0.5%)	6 (0.6%)	0.631	2 (0.5%)	0	0.540	6 (1.0%)	3 (0.5%)	0.341	1 (0.6%)	0	0.496
**Cochlear Implantation**	4 (0.2%)	4 (0.4%)	0.457	1 (0.2%)	3 (1.3%)	0.133	2 (0.3%)	4 (0.7%)	0.687	0	2 (1.2%)	0.499

* Chi-square test or Fisher’s exact test. ^†^ Unclassified severity in 7 and 3 children without and with ROP, respectively, at 18 months and in 1 without ROP at 3 years’ follow-up.

**Table 3 jcm-10-04781-t003:** Clinical factors associated with hearing impairment in univariate and multivariate regression analyses.

Parameters	Impairment at 18 Months				Impairment at 3 Years			
	Univariate		Multivariate		Univariate		Multivariate	
	OR (95% CI)	*p*	OR (95% CI)	*p*	OR (95% CI)	*p*	OR (95% CI)	*p*
Sex	0.58 (0.36–0.93)	0.024	0.62 (0.38–1.01)	0.055	0.46 (0.14–1.51)	0.199	0.50 (0.15–1.67)	0.259
Gestational age	0.91 (0.83–0.99)	0.024	1.02 (0.91–1.14)	0.720	0.80 (0.64–1.0)	0.045	0.72 (0.53–0.98)	0.038
ROP	1.61 (1.01–2.58)	0.045	1.23 (0.68–2.20)	0.490	1.13 (0.37–3.50)	0.832	0.52 (0.12–2.22)	0.375
IVH	1.41 (1.17–1.69)	<0.001	1.30 (1.05–1.60)	0.015	1.10 (0.64–1.88)	0.741	0.90 (0.48–1.66)	0.730
BPD	1.81 (1.12–2.91)	0.015	1.26 (0.71–2.22)	0.427	1.23 (0.78–1.94)	0.371	0.51 (0.11–2.31)	0.385
Sepsis	2.01 (1.22–3.32)	0.006	1.61 (0.94–2.75)	0.082	3.19 (1.02–9.93)	0.045	2.91 (0.83–10.2)	0.097

Abbreviations: OR, odds ratio; CI, confidence interval; ROP, retinopathy of prematurity; IVH, intraventricular hemorrhage; BPD, bronchopulmonary dysplasia.

## Data Availability

The Korean Neonatal Network (KNN) Publication Ethics Policy adheres to the following research data management and access guidelines: All information about patients and participating NICUs is confidential and is only available to individuals who have access for the purposes of the research activities permitted. Access is only allowed for the purpose of collecting data for the first time, and no access for any other purpose is allowed.

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
