# Peer review of "Retinopathy of Prematurity and Hearing Impairment in Infants Born with Very-Low-Birth-Weight: Analysis of a Korean Neonatal Network Database"

_jcm, 2021, doi:10.3390/jcm10204781_

Round 1

Reviewer 1 Report

Overall when p value is mentioned ,I think there is no  need  of mentioning it as significant .As its already stated that p< 0.05 is significant

Abstract: line 21: Writing is as "7940 VLBW infants  who underwent..." is better.

line 27: A higher number of infants did not pass hearing.... (..).With p values mentioned its understood that its significant

Line 152-155: Prevalence of those who did not pass hearing screen is worth mentioning .Infants who passed the hearing screen need not be mentioned

Line 207: Is visual impairment is the outcome of ROP only?As visual impairment can be because of other reasons besides ROP

Line 253 : please write the reference number

217-222: I think its not pertinent to present study

223-224: should be in the methods section

275-277: The statement appears contradictory. With nonsignificant association ,suggesting attention should be paid to hearing function in infants with ROP.

I think it should be presented as explaining more about not significant association of ROP with hearing impairment 

Line 291: Not sure if this conclusion can be drawn that infants with ROP may be at risk of hearing impairment when multivariate analysis doesn't support that

Author Response

Point 1: Overall when p value is mentioned, I think there is no need of mentioning it as significant. As its already stated that p< 0.05 is significant

Response 1: Thank you very much for your meticulous review and valuable suggestions. As per your suggestion, we have deleted the statements regarding statistical significance throughout the manuscript where the p values had already been indicated to be lower than 0.05.

However, as we had already provided the p values in the tables, we have deleted most of the p values mentioned in the text to avoid repetition.

Point 2: Abstract: line 21: Writing is as "7940 VLBW infants who underwent..." is better.

Response 2: Thank you for your suggestion. We have changed the phrase as per your suggestion.

Point 3: line 27: A higher number of infants did not pass hearing.... (..).With p values mentioned its understood that its significant

Response 3: We have changed the sentence to “The frequency of infants who did not undergo hearing screening at near-term ages was higher in the ROP group than in the no-ROP group (18.2% vs. 12.0%, P <0.001), and the prevalence of hearing impairment at 18 months was higher in the ROP group than in the no-ROP group (3.5% vs. 2.2%, P=0.043)” as per your suggestion.

Point 4: Line 152-155: Prevalence of those who did not pass hearing screen is worth mentioning .Infants who passed the hearing screen need not be mentioned

Response 4: The sentence “Among the 7,940 infants who underwent both ROP and hearing screening, 2,028 of 2,480 (81.8%) infants with ROP passed the hearing screening, whereas 4,803 of 5,460 (88.0%) without ROP passed the screening” has been changed to “Among the 7,940 infants who underwent both ROP and hearing screening, 452 of 2,480 infants with ROP failed to pass the hearing screening, whereas 657 of 5,460 infants without ROP did. Accordingly, the prevalence of those who did not pass the hearing screening was 18.2% and 12.0% among the infants with and without ROP, respectively, which was significantly different.” as suggested.

Point 5: Line 207: Is visual impairment is the outcome of ROP only?As visual impairment can be because of other reasons besides ROP

Response 5: We agree with your comment. We could not assess the causes of visual impairment in detail because the specific data for the causes were insufficient, and the causes were not clearly identifiable in some cases. Nevertheless, we addressed both outcomes, as these are major neurosensory outcomes that greatly affect cognitive development in children. In the revised manuscript, we have avoided mentioning that visual impairment in the patients was due to ROP. As the two types of impairments were not significantly associated, we have not addressed the association in detail in our manuscript, although we briefly presented the proportions of hearing impairment in infants with and without visual impairment in the Results section.

Point 6: Line 253 : please write the reference number

Response 6: We apologize for the omission. We have added reference #19 (Schmidt B et al. Association between severe retinopathy of prematurity and nonvisual disabilities at age 5 years. JAMA. 2014 Feb 5;311(5):523-5. doi: 10.1001/jama.2013.282153.) to this sentence.

 Point 7: 217-222: I think its not pertinent to present study

Response 7: We agree with your comment. We have deleted the sentences. To improve the flow of the paragraph, the next sentence has been changed to “Using the data of a large number of VLBW infants registered in a nationwide cohort, our study revealed the frequency of both ROP and hearing impairment and explored the association between them.”

 Point 8: 223-224: should be in the methods section

Response 8: We agree with your comment. However, this has already been mentioned in the Methods section. Accordingly, the sentence has been removed from the Discussion section to avoid repetition.

 Point 9: 275-277: The statement appears contradictory. With nonsignificant association, suggesting attention should be paid to hearing function in infants with ROP.

Response 9: Although the association was not statistically significant, it has been shown that infants with ROP have more frequent hearing impairment at 18 months of corrected age. Although a significant, independent association was not observed, we could suggest that attention should be paid to hearing function in infants with ROP based on the increased frequency of impairment in these infants. This tendency was considered to be due to extreme prematurity in infants with ROP, which may also be associated with impaired hearing function.

 To avoid confusion, we have decided to remove the sentence “Despite the non-significant association between ROP and its treatment and hearing impairment, our results suggest that particular attention should be paid to hearing function in infants with ROP” in response to your comment.

 Point 10: I think it should be presented as explaining more about not significant association of ROP with hearing impairment 

Response 10: Thank you for your valuable comments. In response, we investigated the risk factors for hearing loss and those of ROP. Based on our analyses, the risk factor common to both hearing loss and ROP was extreme prematurity. This explains why a non-significant association was noted after controlling for gestational age and prematurity-associated comorbidities in our study. The common pathogenic background is considered to be mainly mediated by extreme prematurity, and the mechanisms of hearing impairment and ROP seem to be independent of each other.

Accordingly, we have added a new paragraph “Risk factors for ROP suggested by previous studies include maternal factors (maternal diabetes mellitus and hypertension), prenatal and perinatal factors (chorioamnionitis and premature rupture of membrane), infant factors (low gestational age [23], small birth weight [23], twin birth [24], and low Apgar score [25]), medical interventions (oxygen [26], respiratory support [27]), comorbidities of prematurity (BPD [28], IVH [29], NEC [30], and sepsis [29]), and genetic factors [31,32]. Those of childhood hearing loss contain preterm birth, intracranial hemorrhage, the use of ototoxic medications, congenital cytomegalovirus, toxoplasmosis, admission to a NICU, and a family history of hearing loss.[3,33] Extreme prematurity and associated comorbidities were the common risk factors for both ROP and hearing loss. Therefore, after controlling for extreme prematurity, the association between ROP and hearing loss was shown to be nonsignificant, and, mechanistically, the functional development of hearing might not be directly related to the pathologic processes (i.e. retinal vasoproliferation) occurring in ROP.”

 Point 11: Line 291: Not sure if this conclusion can be drawn that infants with ROP may be at risk of hearing impairment when multivariate analysis does n't support that

Response 11: As per your suggestion, we have changed the sentence “Our findings suggest that infants with ROP may be at risk of hearing impairment in early developmental periods” to “Among VLBW infants, the frequency of hearing impairment at 18 months of age was higher in those with ROP than in those without ROP” to avoid confusion but directly restate our main result.

Reviewer 2 Report

I totally agree with the authors that the overlap of visual and hearing impairments has a significant risk on development of the infants. However, it seems more natural to think that visual impairment is not associated with hearing impairment, but that birth and growth conditions affect the development of these function. In this paper, the relationship with hearing loss is analyzed based on ROP, but it is also possible to investigate the relationship with ROP based on the presence or absence of hearing loss. This analysis method seems to be unnatural. In addition, the analysis method must be corrected by considering many confounding factors, such as birth weight, gestation of weeks, or complications of the patients and so on. Therefore, I think that the analysis strategy of this paper is wrong. For example, table 1 compares various elements with and without ROP, but I think this analysis should be done including the covariate factor (for example, propensity score analysis). Moreover, I think that the result that ROP and hearing loss are not related in the multivariate analysis in table3 clearly shows the reason why ROP and hearing loss should not be statistically connected.
However, the results of such a large-scale survey are invaluable. How about simply showing the proportion of ROP and hearing impairment, or the relationship between visual/hearing impairment and development-related factors ?

Author Response

Response to Reviewer 2 Comments

Point 1: I totally agree with the authors that the overlap of visual and hearing impairments has a significant risk on development of the infants. However, it seems more natural to think that visual impairment is not associated with hearing impairment, but that birth and growth conditions affect the development of these function. In this paper, the relationship with hearing loss is analyzed based on ROP, but it is also possible to investigate the relationship with ROP based on the presence or absence of hearing loss. This analysis method seems to be unnatural. In addition, the analysis method must be corrected by considering many confounding factors, such as birth weight, gestation of weeks, or complications of the patients and so on. Therefore, I think that the analysis strategy of this paper is wrong.

Response 1: We appreciate your comments and helpful suggestions. We agree that considering many confounding factors is important to explore the true association between ROP and hearing loss. However, ROP mostly occurred before the term-equivalent periods, and hearing impairment, as the main outcome, was assessed later (18 months and 3 years). In this prospective study with a large sample size, we attempted to address the differences in functional outcome (hearing impairment) at early developmental periods between very-low-birthweight infants with and without ROP having been determined before discharge from the neonatal intensive care unit of participating centers (mostly at term-equivalent periods).

   As suggested by the reviewer, it is possible to investigate the relationship between the functional outcome and disease occurring much earlier than the outcome based on the presence/absence of hearing loss. This is mostly performed through retrospective cohort studies looking back at whether the risk was different between the presence/absence of the outcome variable. However, our prospective cohort study could address how frequent the infants with ROP would have hearing impairment, which is of concern to ophthalmologists and pediatricians regarding their VLBW patients with ROP (unsure whether they will develop hearing impairment or not) and even those without ROP.

   As noted in previous studies, ROP occurs in premature babies with more frequent prematurity-associated conditions. These may also be associated with hearing outcomes, which should be controlled for to investigate the association between ROP and hearing outcome. To control for possible confounders, we included prematurity and prematurity-associated medical conditions in the multivariate regression analyses. However, given the significant correlation between gestational age and birthweight, we confronted the problem of collinearity in these analyses, which reduced the precision of the estimated coefficients and weakened the statistical power of the regression model. Thus, we excluded birthweight but included gestational age in our multivariate analyses to avoid the collinearity issues for such highly correlated variables. We added several prematurity-associated factors and conditions if these were significantly associated with ROP or hearing impairment to minimize confounding bias. Our group has published neurodevelopmental outcomes and white matter abnormalities in infants with and without ROP using the same methods (References #15 and #16; Ahn et al. Brain White Matter Maturation and Early Developmental Outcomes in Preterm Infants With Retinopathy of Prematurity. Invest Ophthalmol Vis Sci. 2021 Feb 1;62(2):1-9. and Ahn et al. Diffusion Tensor Imaging Analysis of White Matter Microstructural Integrity in Infants With Retinopathy of Prematurity. Invest Ophthalmol Vis Sci. 2019 Jul 1;60(8):3024-3033).

   However, we agree with the reviewer’s point for the stricter method (e.g., propensity score matching) to control for potential confounding variables. In this revision, we have added the analyses obtained from propensity score matching (Tables 1 and 2) As also mentioned by the reviewer (Point #3), we believe that the overall data from a large number of VLBW infants are valuable and informative, and we have not deleted the results obtained from our analyses of the overall data. Accordingly, the two different methods for adjusting prematurity-related variables, multivariate analyses and propensity score matching, showed no significant association between ROP and hearing impairment. We believe that the implementation of the reviewer’s suggestions has greatly improved the overall presentation and precision of our analyses. Again, thank you very much for your valuable suggestions.

Point 2: For example, table 1 compares various elements with and without ROP, but I think this analysis should be done including the covariate factor (for example, propensity score analysis). Moreover, I think that the result that ROP and hearing loss are not related in the multivariate analysis in table3 clearly shows the reason why ROP and hearing loss should not be statistically connected.

Response 2: As mentioned in our response to Point #1, we performed multivariate regression analyses using gestational ages and prematurity-associated conditions as covariates. However, as suggested, propensity score analyses were performed and have been added to this revision.

Point 3: However, the results of such a large-scale survey are invaluable. How about simply showing the proportion of ROP and hearing impairment, or the relationship between visual/hearing impairment and development-related factors ?

Response 3: As per your suggestion, we have added Supplementary Table 1 summarizing the proportion of ROP and hearing impairment assessed at the two developmental periods in our cohort. The relationship between visual and hearing impairment in our cohort has been addressed in Section 3.3 and Supplementary Table 2. To evaluate the association between the two types of impairment by controlling for development-related factors, we used multivariate logistic regression analyses and identified no significant association between visual and hearing impairment.

  However, visual and hearing impairment and their association have been investigated by multiple studies, and our focus was not on the relationship between visual and hearing impairment, the two major neurosensory outcomes. As the main purpose of this study was hearing impairment in infants with ROP (and also that in those without ROP) and an independent association between ROP and hearing impairment, we have included both our previous results and those suggested by the reviewers.

Round 2

Reviewer 2 Report

This study has a very large number of subjects, and I think it is useful data.